# Conformal Graph-level Out-of-distribution Detection with Adaptive Data Augmentation

## ABSTRACT

Graph-level out-of-distribution (OOD) detection, which attempts to identify OOD graphs originated from an unknown distribution, is a vital building block for safety-critical applications in Web and society. Current approaches concentrate on how to learn better graph representations, but fail to provide any statistically guarantee on detection results, therefore impeding their deployments in the scenario where detection errors would result in serious consequences. To overcome this critical issue, we propose the **C**onformal **G**raph-level **O**ut-of-distribution **D**etection (CGOD), extending the theory of conformal prediction to graph-level OOD detection with a rigorous control over the false positive rate. In CGOD, we develop a new aggregated non-conformity score function based on the proposed adaptive data augmentation. Through the guidance from two designed metrics, i.e., score consistency and representation diversity, our augmentation strategy can generate multiple non-conformity scores, and aggregating these generated non-conformity scores together is robust to the misleading information. Meanwhile, our score function can perceive the subsequent process of conformal inference, enabling the aggregated non-conformity score to be adaptive to different input graphs and deriving a more accurate score estimation. We conduct experiments on multiple real-world datasets with different empirical settings. Extensive results and model analyses demonstrate the superior performance of our approach over several competitive baselines.

## CCS CONCEPTS

• **Computing methodologies → Neural networks**; • **Information systems → Data mining**.

## KEYWORDS

Graph-level out-of-distribution detection, conformal prediction, graph neural networks

**ACM Reference Format:**
Anonymous Author(s). 2018. Conformal Graph-level Out-of-distribution Detection with Adaptive Data Augmentation. In *Proceedings of Make sure to enter the correct conference title from your rights confirmation emai (Conference acronym 'XX)*. ACM, New York, NY, USA, 11 pages. https://doi.org/XXXXXXX.XXXXXXX

## 1 INTRODUCTION

Graph-level applications are ubiquitous in diverse set of research fields, such as social network analysis [22, 27, 41], molecular property prediction [6, 33, 40] and intelligent traffic forecasting [9, 13, 39]. Recently, a significant amount of approaches for these graph-level applications have been proposed [4, 44, 50], among which graph neural networks (GNNs) have received a great deal of attention [11, 16, 35]. Typically, GNNs are trained with a closed-world assumption that training graphs and test graphs follow the same data distribution [5, 20, 42]. Nevertheless, many real-world situations violate this closed-world assumption and, instead, involve out-of-distribution (OOD) graphs which have not been seen during the training process [8, 21, 37]. In fact, for an ideal graph machine learning model, it should not only make predictions on in-distribution (ID) graphs correctly, but also be able to detect such OOD samples during the inference phase to mitigate some unexpected risks, when being deployed on practical scenarios [14, 48].

On the basis of this high security demand, **graph-level OOD detection**, which aims to determine whether an input graph is ID or OOD, has become an important research direction. Previous works adopt different learning paradigms, e.g., contrastive learning [28] and prompt learning [23] to learn effective ID features, with the goal of enlarging the score gap between ID and OOD graphs. For example, GOOD-D [24] designs a method of hierarchical contrastive learning to capture the common patterns of ID graphs in different granularities (node-, graph- and group-levels) so that OOD graphs which violate these patterns can be easily exposed. AAGOD [10] proposes a post-hoc method, enabling a pre-trained GNN to detect OOD graphs without modifying model parameters. It leverages a graph prompt on the adjacency matrix to amplify the structural difference between ID and OOD graphs.

Despite their promising results, existing works of graph-level OOD detection remain at designing more advanced models, but fail to provide any statistically guarantee on detection results. Such a lack of rigor impedes their applications in the scenario where detection errors would result in serious consequences. In this paper, we attempt to establish the connection between graph-level OOD detection and conformal prediction (CP) [36] for filling this important gap. CP is a useful tool of generating prediction sets with a coverage guarantee that such sets cover the true label with a user-specified threshold. To be specific, we propose the **C**onformal **G**raph-level **O**ut-of-distribution **D**etection named **CGOD**, a novel framework that extends CP to graph-level OOD detection with a rigorous control over the false positive rate (FPR). Additionally, CGOD is a flexible and light-weighted architecture which can be paired with different detection models to identify OOD graphs in a post-processing manner.

The non-conformity score function, which quantifies how different the input graph is from the training distribution, is the primary

factor for CP [1]. However, when handling graph-level OOD detection, it suffers from two critical limitations: (1) It follows a single point estimation of the non-conformity score, which is easily affected by the misleading information including the high estimation variance and the noise feature of recognizing OOD characteristics; (2) It is a pre-determined function, which cannot make adaptive adjustments to perceive the subsequent estimation process, leading to an inaccurate calculation. In this paper, we develop a new **aggregated non-conformity score function** to overcome above limitations, based on the proposed adaptive data augmentation.

Specifically, our method *differentiates* the process of conformal inference. Through optimizing two designed metrics, i.e., score consistency and representation diversity, CGOD can utilize our augmentation strategy to generate multiple non-conformity scores. By aggregating these scores together, CGOD derives an aggregated non-conformity score to enhance the robustness of the misleading information. Furthermore, due to the fact that our score function is trainable, it can be aware of the calculation of the non-conformity score, enabling the aggregated non-conformity score to be adaptive to different input graphs and deriving a more accurate score estimation. In general, the main contributions of this paper are summarized here:

- To the best of our knowledge, we are the first work that attempts to formulate the task of graph-level OOD detection from the perspective of CP, facilitating a rigorous control over the FPR while ensuring detection performance.
- We design two novel metrics to guide the proposed adaptive data augmentation to generate multiple non-conformity scores leading to an aggregated non-conformity score function for boosting model performance effectively.
- Experiments are conducted on multiple real-world datasets in different empirical settings. Extensive results and detailed analyses validate the superiority of our method over multiple strong counterparts[1].

## 2 RELATED WORK

### 2.1 Conformal Prediction on Graphs

CP [1, 36] is an uncertainty quantification framework which can produce statistically valid prediction sets (or intervals) for any pre-trained machine learning models, only assuming exchangeability of the data. The basic idea of CP is to estimate the $p$-value for each possible label of a new sample and exclude from the prediction set those labels having a $p$-value less than a user-specified threshold $\epsilon$. Similar to statistical hypothesis testing [15], CP aims to reject the most unlikely labels at significance level $\epsilon$. For estimating these $p$-values, CP leverages *the non-conformity score function* to measure how different a given sample is relative to a set of training samples. Thus, the non-conformity score function plays a crucial role of determining the usefulness of CP.

CP has proven effective in numerous domains. For example, ICAD [18, 19] exploits CP to detect anomalous trajectories. Recent works investigate how to use CP to solve problems on graphs. DAPS [47] and CF-GNN [12] study the exchangeability of GNNs under the semi-supervised node classification, and propose different

---

[1]The source code is available at https://anonymous.4open.science/r/CGOD/

**Table 1: Used notations.**

| Notation | Description |
|---|---|
| $\mathbb{P}^{in}, \mathbb{P}^{out}$ | ID and OOD distributions of graphs |
| $\mathcal{D}^{in}, \mathcal{D}^{out}$ | ID and OOD datasets |
| $\mathcal{D}_{tr}, \mathcal{D}_{te}$ | training and test datasets |
| $\mathcal{D}_{ptr}, \mathcal{D}_{cal}$ | proper training and calibration dataset |
| $\mathcal{D}_{cal-tr}, \mathcal{D}'_{cal}$ | trainable and remaining calibration datasets |
| $G^{in}$ and $G_{te}$ | a ID graph from $\mathcal{D}^{in}$ and a test graph from $\mathcal{D}_{te}$ |
| $\text{detect}(\cdot)$ | detection function |
| $F(\cdot, \cdot)$ | transformation function |
| $\Phi(\cdot)$ | a GNN encoder |
| $V(\cdot, \cdot)$ and $s$ | non-conformity score function with its score |
| $\Psi^{j,1:k}$ | $k$ data augmentations of the $j$-th graph |
| $\widehat{V}(\cdot, \cdot; \cdot)$ | aggregated non-conformity score function |
| $\hat{s}$ | aggregated non-conformity score |
| $\boldsymbol{g}_j, \hat{\boldsymbol{g}}_j^i$ | graph representation with its $i$-th augmentation |
| $p_{te}, \hat{p}_{te}$ | $p$-value and aggregated $p$-value |

strategies to improve efficiency. CoDrug [17] introduces conformal molecular graph prediction, which adopts kernel density estimation to handle the problem of covariate shift. Different from above approaches, our method extends CP to graph-level OOD detection and develops a new aggregated non-conformity score function to improve model effectiveness.

### 2.2 OOD Detection on Graphs

OOD detection [29] attempts to identify OOD samples from ID data, which is an essential problem for deploying machine learning models on safety-critical applications in social networks. Many methods have studied the problem of OOD detection on graphs [43]. Among them, GOOD-D [24] is the first work focusing on graph-level OOD detection. As described in Introduction, GOOD-D designs a self-supervised method that contrasts different granularities to capture the ID patterns from both feature and structure views, so as to detect OOD graphs based on the discrepancy in these granularities. GOODAT [38] is one of follow-up works, which uses the information bottleneck to capture informative sub-graphs for achieving OOD detection in test time.

Graph-level anomaly detection [26] is a sub-area of graph-level OOD detection, since anomaly or malicious graphs can be regarded as a certain type of OOD data. There are many promising works on graph-level anomaly detection. For instance, OCGIN [49] is an end-to-end model which adopts GNNs to learn graph representations simultaneously optimizes an anomaly detection objective, e.g., one-class classification or reconstruction loss function. GLocalKD [25] learns global- and local-sensitive graph normality to detect anomalous graphs by the joint random distillation of graph and node representations. However, all above works fall short in providing any statistically guarantee on detection results.

## 3 PRELIMINARY

In this section, we first provide the preliminary in our paper and the used notations are summarized in Table 1. Let $G = (\mathcal{V}, \mathcal{E}, \mathbf{X})$ denote an undirected graph, and $\mathcal{V}$ and $\mathcal{E}$ represent the sets of

nodes and edges. $\mathbf{X} \in \mathbb{R}^{|\mathcal{V}| \times d}$ represents the feature matrix, where $d$ is the feature dimension. $\mathbf{A} \in \mathbb{R}^{|\mathcal{V}| \times |\mathcal{V}|}$ is the adjacency matrix and each element $\mathbf{A}_{i,j} \in \{0, 1\}$ denotes the connectivity between nodes $i$ and $j$. Following the previous work [24], the definition of graph-level OOD detection is given as follows,

DEFINITION 1 (GRAPH-LEVEL OOD DETECTION). *Assuming that we have an ID dataset $\mathcal{D}^{in}$ where each input graph is originated from the distribution $\mathbb{P}^{in}$ and an OOD dataset $\mathcal{D}^{out}$ where each input graph is originated from the distribution $\mathbb{P}^{out}$. The training dataset $\mathcal{D}_{tr} = \{G_1^{in}, \ldots, G_n^{in}\}$ is a subset of $\mathcal{D}^{in}$ which only includes ID graphs ($n$ is the size of $\mathcal{D}_{tr}$), while the test dataset $\mathcal{D}_{te}$ is constituted by two separated datasets $\mathcal{D}_{te}^{in} \subset \mathcal{D}^{in}$ and $\mathcal{D}_{te}^{out} \subset \mathcal{D}^{out}$, i.e., $\mathcal{D}_{te} = \mathcal{D}_{te}^{in} \cup \mathcal{D}_{te}^{out}$ and $\mathcal{D}_{tr} \cap \mathcal{D}_{te}^{in} = \emptyset$. For an arbitrary test sample $G_{te} \in \mathcal{D}_{te}$, the goal of graph OOD detection is to distinguish which distribution ($\mathbb{P}^{in}$ or $\mathbb{P}^{out}$) $G_{te}$ belongs to.*

Graph-level OOD detection typically follows an *unsupervised* setting in which only unlabeled ID graphs are available for identifying OOD graphs. Most current approaches focus on designing an effective GNN encoder $\Phi(\cdot)$ to learn the graph representation $\mathbf{g} \in \mathbb{R}^c$, i.e., $\mathbf{g} = \Phi(G)$, where $c$ denotes the representation dimension. Afterwards, a transformation function $F(\cdot, \cdot)$ that transforms the graph representation into the OOD detection score can be applied on $\mathbf{g}$ to identify its OOD-ness:

$$\text{detect}(G) = \begin{cases} 1 & F(\Phi(G), \mathcal{D}_{tr}) \geq \gamma \\ 0 & F(\Phi(G), \mathcal{D}_{tr}) < \gamma. \end{cases} \quad (1)$$

Here 0 and 1 indicate the ID sample and the OOD sample, respectively. SSD [31] is a well-known transformation function for OOD detection. It uses the $k$-means algorithm to group graph representations into $T$ clusters and adopts the Mahalanobis distance between $\mathbf{g}$ and the nearest cluster center as the OOD detection score:

$$F(\Phi(G), \mathcal{D}_{tr}) = F(\mathbf{g}, \mathcal{D}_{tr}) = \min_t (\mathbf{g} - \boldsymbol{\mu}_t)^\top \Sigma_t^{-1} (\mathbf{g} - \boldsymbol{\mu}_t), \quad (2)$$

$\boldsymbol{\mu}_t$ and $\Sigma_t$ are the sample mean and the sample covariance of the cluster $t$ generated from $\mathcal{D}_{tr}$.

## 4 METHODOLOGY

In this section, we first delve into how to achieve graph-level OOD detection from the view of CP. This process follows a post-processing manner that can be armed with different pre-trained detection models to achieve prediction. We then introduce the key component of CGOD: the aggregated non-conformity score function based on the proposed adaptive data augmentation. Figure 1 shows a sketch of CGOD. Model analysis including the theoretical detection guarantee and time complexity is given in the end.

### 4.1 Model Overview

CGOD follows a most widely used case of CP, i.e., split conformal prediction (SCP) [36] for model implementation. Specifically, CGOD includes the following three steps: (1) **Data split**. For improving computational efficiency, CGOD first splits $\mathcal{D}_{tr}$ into a *proper training* set $\mathcal{D}_{ptr} = \{G_j^{in}\}_{j=1}^m$ and a *calibration* set $\mathcal{D}_{cal} = \{G_j^{in}\}_{j=m+1}^n$, i.e., $\mathcal{D}_{tr} = \mathcal{D}_{ptr} \cup \mathcal{D}_{cal}$. (2) **Non-conformity score function**. Given an arbitrary graph $G$, CGOD has to define the non-conformity score function $V(G, \mathcal{D}_{ptr}) = s$ for testing whether $G$ conforms to

$\mathcal{D}_{ptr}$, where $s$ indicates how different $G$ is relative to the graph samples in $\mathcal{D}_{ptr}$. A higher $s$ demonstrates that $G$ is more different from $\mathcal{D}_{ptr}$. To be consistent with Eq.(1-2), we can set

$$V(G, \mathcal{D}_{ptr}) := F(\Phi(G), \mathcal{D}_{ptr}), \quad (3)$$

where $\Phi(\cdot)$ and $F(\cdot, \cdot)$ can be pre-trained models. (3) **$P$-value estimation**. The non-conformity score function then can be applied to the calibration set $\mathcal{D}_{cal}$ for generating the non-conformity score $s_j = V(G_j^{in}, \mathcal{D}_{ptr})$ of each graph $G_j^{in} \in \mathcal{D}_{cal}$. Likewise, the non-conformity score of a test sample $G_{te}$ can be calculated as $s_{te}$. Building upon these non-conformity scores, the $p$-value of $G_{te}$ is estimated as the ratio of $s_{m+1}, \ldots, s_n$ that are at least as large as $s_{te}$:

$$p_{te} = \frac{|\{j = m + 1, \ldots, n : s_j \geq s_{te}\}| + 1}{n - m + 1}. \quad (4)$$

When $p_{te}$ is smaller than a given OOD threshold $\epsilon \in (0, 1)$, $G_{te}$ is identified as an OOD sample. As suggested by previous works [2], the above process can be viewed as a *statistical hypothesis testing* where the null hypothesis is that $G_{te}$ is originated from the same distribution as the graph samples in $\mathcal{D}_{ptr}$, and $p_{te}$ can be interpreted as the probability of rejecting this null hypothesis wrongly.

Furthermore, applying CP to graph-level OOD detection requires a critical assumption that the condition of exchangeability[2] holds for graph data [1, 12, 47]. We highlight that CGOD can satisfy exchangeability via meeting the following *independence* proposition:

PROPOSITION 1. *In the setting of graph-level OOD detection described in Definition 1, the non-conformity score function $V(\cdot, \cdot)$, the calibration set $\mathcal{D}_{cal}$ and the test dataset $\mathcal{D}_{te}$ are independent of each other.*

The corresponding proof is evident: the non-conformity score function $V(\cdot, \cdot)$ is built upon the proper training set $\mathcal{D}_{ptr}$. $\mathcal{D}_{ptr}$ and $\mathcal{D}_{cal}$ are i.i.d. Meanwhile, $\mathcal{D}_{te}$ is another independent fold of the used graph data.

### 4.2 Aggregated Non-conformity Score Function

According to the above three steps, the non-conformity score function is the primary factor in CP, because it incorporates almost all the information for determining whether the given graphs are OOD samples. However, it still exists two critical limitations as described in Introduction: First, $V(\cdot, \cdot)$ follows a single point estimation so that it is easily affected by the misleading information; Second, $V(\cdot, \cdot)$ is a pre-determine function which cannot perceive the subsequent process of $p$-value estimation.

To overcome these limitations, *we differentiate the process of conformal inference and introduce a novel aggregated non-conformity score function*. In our score function, we propose the adaptive data augmentation where each data augmentation captures different representation diversities and corresponds to a non-conformity score. Aggregating multiple non-conformity scores together can reduce the high variance introduced by the single point estimation and is robust to the noise feature of recognizing OOD characteristics. Meanwhile, the proposed augmentation strategy is trainable, so our score function can be aware of the follow-up $p$-value estimation,

---

[2]Exchangeability. This condition requires the distribution to be invariant to permutations of the elements in $\{G_1, \ldots, G_n\}$. More precisely, for each finite $n$, if $\pi$ is a permutation of $\{1, \ldots, n\}$, then: $\mathbb{P}(G_{\pi(1)}, \ldots, G_{\pi(n)}) = \mathbb{P}(G_1, \ldots, G_n)$.

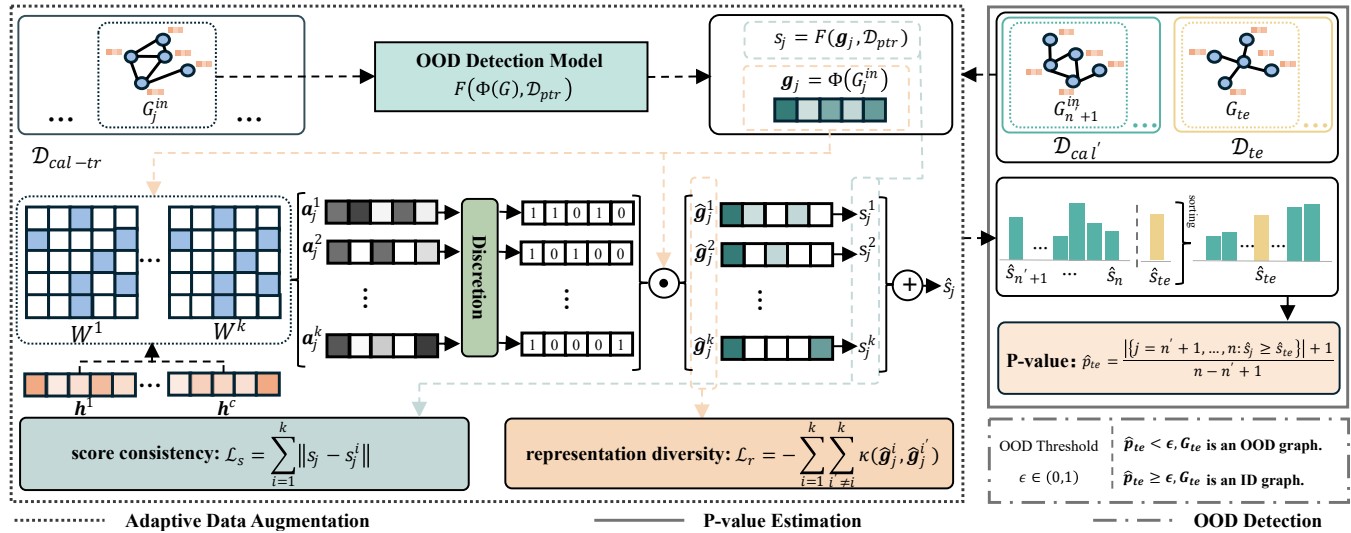

**Figure 1: CGOD for graph-level OOD detection. In the training phase, we design the metrics of score consistency and representation diversity to optimize our proposed adaptive data augmentation. In the inference phase, CGOD first calculates these aggregated non-conformity scores of $\mathcal{D}'_{cal}$ and $G_{te} \in \mathcal{D}_{te}$, and then derives the $p$-value of $G_{te}$ for achieving OOD detection.**

enabling the aggregated non-conformity score to be adaptive to different input graphs and thereby deriving a more accurate score estimation.

Specifically, given $k$ data augmentations $\Psi^{1:k} = (\Psi^1, \ldots, \Psi^k)$, CGOD first generates $k$ non-conformity scores $s^{1:k} = (s^1, \ldots, s^k)$ for $G$:

$$s^{1:k} = \left(V(\Psi^1(G), \mathcal{D}_{ptr}), \ldots, V(\Psi^k(G), \mathcal{D}_{ptr})\right). \quad (5)$$

Afterwards, our score function $\widehat{V} = (G, \mathcal{D}_{ptr}; \Psi^{1:k})$ is set as an element-wise strictly increasing function with the following aggregated non-conformity score:

$$\hat{s} = \widehat{V}(G, \mathcal{D}_{ptr}; \Psi^{1:k}) = \sum_{i=1}^{k} V(\Psi^i(G), \mathcal{D}_{ptr}). \quad (6)$$

We explain why use this function setting in Model Analysis. In the next, we describe how to optimize these data augmentations.

*Adaptive Data Augmentation.* Instead of using some heuristic augmentations working on graph structures and node features [30], our augmentation strategy follows a trainable fashion which perturbs learned graph representations in the embedding space. This approach has been demonstrated to be a simple yet effective way of introducing noise to graphs [3]. To train these data augmentations, we first split a small number of data from $\mathcal{D}_{cal}$ as the trainable calibration set $\mathcal{D}_{cal-tr} = \{G_j^{in}\}_{j=m+1}^{n'}$, and the remaining data $\mathcal{D}'_{cal} = \{G_j^{in}\}_{j=n'+1}^{n}$ is still used as the calibration set, i.e., $\mathcal{D}_{cal} = \mathcal{D}_{cal-tr} \cup \mathcal{D}'_{cal}$. We then design two metrics, i.e., score consistency and representation diversity as the loss function to train these data augmentations on $\mathcal{D}_{cal-tr}$.

For an arbitrary graph $G_j^{in} \in \mathcal{D}_{cal-tr}$, each data augmentation would generate an augmented graph representation $\hat{g}_j^i = \Psi^{j,i}(g_j; \theta^i)$, where $g_j$ denotes the original graph representation of

$G_j^{in}$ and $\theta^i$ denotes the model parameters. We parameterize each data augmentation $\Psi^{j,i}$ as a masking operation that would mask some inessential or redundant dimensions. To effectively determine which dimensions should be masked, we first use the attention mechanism $\text{att}(g_j; \theta^i)$ to derive a weighted vector $a_j^i \in \mathbb{R}^c$ where each element $a_j^{i,z}$ represents the importance of $z$-th dimension:

$$a_j^{i,z} = \frac{\exp(\text{LeakyReLU}(g_j^{\mathsf{T}} W^i h^z))}{\sum_{o=1}^{c} \exp(\text{LeakyReLU}(g_j^{\mathsf{T}} W^i h^o))}. \quad (7)$$

Here $W^i$ is a parameter matrix in $\Psi^{j,i}$, and $\{h^o\}_{o=1}^{c}$ represents a set of dimension-wise trainable vectors. We then perform a discretization operation on $a_j^i$, i.e., $\text{disc}(a_j^i; \xi)$ where the highest values are discretized as ones and the others are discretized as zeros, with a specific ratio $\xi$ for controlling the proportion of zeros. $\hat{g}_j^i$ is finally defined as

$$\hat{g}_j^i = \Psi^{j,i}(g_j; \theta^i) = \text{disc}(\text{att}(g_j; \theta^i); \xi) \odot g_j, \quad (8)$$

where $\odot$ denotes the element-wise multiplication. Once $\hat{g}_j^i$ has been generated, we can further derive its corresponding non-conformity score $s_j^i = F(\hat{g}_j^i, \mathcal{D}_{ptr})$.

*Designed Metrics.* We design two metrics, i.e., score consistency and representation diversity to train our augmentation strategy, with the goal of improving the robustness of the noise feature of recognizing OOD characteristics. Concretely, given $k$ augmented graph representations $\hat{g}_j^{1:k} = (\hat{g}_j^1, \ldots, \hat{g}_j^k)$, the score consistency requires that their corresponding non-conformity scores $s_j^{1:k} = (s_j^1, \ldots, s_j^k)$ are consistent with the original non-conformity score $s_j$ of the input graph $G_j^{in}$. Meanwhile, representation diversity encourages $\hat{g}_j^{1:k}$ to be dissimilar from each other. Combining these two metrics together, we can define the following loss function:

$$\mathcal{L} = \mathcal{L}_s + \lambda \mathcal{L}_r = \underbrace{\sum_{i=1}^{k} ||s_j - s_j^i||}_{\text{score consistency}} - \lambda \underbrace{\sum_{i=1}^{k} \sum_{i' \neq i}^{k} \kappa(\hat{g}_j^i, \hat{g}_j^{i'})}_{\text{representation diversity}}, \quad (9)$$

where $\mathcal{L}_s$ and $\mathcal{L}_r$ denote the loss functions of score consistency and representation diversity, $\lambda$ is a harmonic factor, and $\kappa$ represents a distance function for measuring the distance between two graph representations. In fact, each non-conformity score corresponds to a $p$-value:

$$p_j^i = \frac{|\{j' = n' + 1, \dots, n : s_{j'} \geq s_j^i\}| + 1}{n - n' + 1}. \quad (10)$$

But directly optimizing the consistency of the $p$-values generated by these augmented graph representations is non-differentiable due to Eq.(10). So we minimize the loss function of score consistency to ensure $p$-value consistency for overcoming the second limitation mentioned in Section 4.2.

Overall, through optimizing these two metrics, CGOD can be aware of the downstream of conformal inference, generating the augmented graph representations that own a high-level consistency of $p$-value estimation with the original graph representation while incorporating as much representation diversity as possible. Moreover, our proposed adaptive data augmentation is not limited to the graph-level representation. It is a general augmentation strategy, which can be armed with different level representations, e.g., the sub-graph level to calculate the corresponding non-conformity scores. Due to this property, CGOD can be combined with different detection models to recognize OOD graphs.

## 4.3 Model Analysis

### 4.3.1 Detection Guarantee.
According to the standard principle of CP [7, 36], these generated $k$ non-conformity scores are required to be exchangeable random variables. To approximate this condition, we parameterize a set of graph data augmentations $\mathcal{S}$ via learned graph representations and the attention mechanism as described in Eq.(7). In this way, each data augmentation can be regarded as one data point which is randomly and independently sampled from $\mathcal{S}$. For avoiding ties, we assume that these aggregated non-conformity scores of an arbitrary test sample and of the samples in $\mathcal{D}_{cal-tr}$ are distinct with probability 1. To comply with this assumption, we set the aggregated non-conformity score function as an element-wise strictly increasing function, and suppose that the data distribution is absolutely continuous w.r.t Lebesgue measure.

THEOREM 4.1. *Given that $\mathcal{S}$ represents a set of graph data augmentations, the aggregated non-conformity score of an arbitrary graph $G_j^{in} \in \mathcal{D}'_{cal}$ is denoted as*

$$\hat{s}_j = \widehat{V}(G_j^{in}, \mathcal{D}_{ptr}; \Psi^{j,1:k}) = \sum_{i=1}^{k} V(\Psi^{j,i}(G_j^{in}), \mathcal{D}_{ptr}), \quad (11)$$

*where $\Psi^{j,i}$ is sampled from $\mathcal{S}$ independently. The aggregated non-conformity score $\hat{s}_{te}$ of the test graph $G_{te}$ follows the same computational procedure with $\hat{s}_j$. If $G_{te}$ belongs to the distribution $\mathbb{P}^{in}$, then*

its $p$-value is defined as

$$\hat{p}_{te} = \frac{|\{j = n' + 1, \dots, n : \hat{s}_j \geq \hat{s}_{te}\}| + 1}{n - n' + 1}, \quad (12)$$

which follows a discrete uniform distribution:

$$\hat{p}_{te} \sim Uniform\left(\left\{\frac{1}{n - n' + 1}, \frac{2}{n - n' + 1}, \dots, 1\right\}\right). \quad (13)$$

Here we denote the new $p$-value of $G_{te}$ as the aggregated $p$-value, i.e., $\hat{p}_{te}$ to distinguish from the original $p$-value $p_{te}$ in Eq.(4). The corresponding proof has been provided in Appendix A.1. If $\hat{p}_{te}$ is smaller than a given OOD detection threshold $\epsilon$, then $G_{te}$ is recognized as an OOD sample. Based on Theorem 4.1, we have the following detection guarantee:

THEOREM 4.2. *Suppose the calibration set $\mathcal{D}'_{cal}$ sampled from the distribution $\mathbb{P}^{in}$, the test dataset $\mathcal{D}_{te}$ and the aggregated non-conformity score function $\widehat{V}(\cdot, \cdot; \cdot)$ are independent of each other. Given a test graph $G_{te} \in \mathcal{D}_{te}$ with its aggregated $p$-value $\hat{p}_{te}$ calculated by Eq.(12) and a specified OOD threshold $\epsilon$, then we have the following upper bound:*

$$Pr(\hat{p}_{te} < \epsilon | G_{te} \sim \mathbb{P}^{in}) \leq \epsilon. \quad (14)$$

Theorem 4.2 demonstrates that our proposed aggregated non-conformity score function can control the type I error rate guaranteeing a bounded FPR for graph-level OOD detection. The corresponding proof is provided in Appendix A.2.

### 4.3.2 Time Complexity.
Appendix A.3 shows the training and inference pseudo-codes of CGOD. CGOD is a light-weighted architecture, which adopts the batch manner for model training and inference. The time complexity of model training is $O(B(|\mathcal{D}'_{cal}|k(c^2 + c) + k^2 + c))$, where $B$ denotes the batchsize. In the inference phase, we can calculate these aggregated non-conformity scores of $\mathcal{D}'_{cal}$ in advance, which largely reduces the time complexity. Based on this operation, the time complexity of model inference is $O(B(k(c^2 + c) + |\mathcal{D}'_{cal}|))$. Since the above parameters typically take small values, so CGOD keeps good model efficiency. In Appendix A.8, we provide the concrete runtime comparison.

## 5 EXPERIMENTS

In this section, we conduct extensive experiments to answer the following research questions (RQs):

- **RQ1**: Can our method be combined with different detection models to improve empirical performance?
- **RQ2**: Can our method provide a rigorous control over the false positive rate?
- **RQ3**: Does our method achieve the supreme performance in comparison with multiple strong baselines?
- **RQ4**: What are the contributions of the proposed different components in our method?
- **RQ5**: How sensitive is our method with respect to different hyper-parameters?

To answer these above questions, we conduct a detailed comparative analysis.

**Table 2: OOD performance comparison (%) of adapting different detection methods ($GIN_D$, $IG_D$ and GOOD-D) to our framework on six ID & OOD datasets.**

| ID | OOD | Metric | $GIN_D$ | $GIN_D$+CP | $CGOD_S$ | $IG_D$ | $IG_D$+CP | $CGOD_U$ | GOOD-D | GOOD-D+CP | $CGOD_E$ |
|---|---|---|---|---|---|---|---|---|---|---|---|
| IMDB-M | IMDB-B | AUC | 74.43 | 74.08 | **77.07** | 74.39 | 74.27 | **75.81** | 77.40 | 77.06 | **79.02** |
| | | AUPR | 65.05 | 64.32 | **69.08** | 68.08 | **68.12** | 66.78 | 67.84 | 70.29 | **71.95** |
| | | FPR95 | 69.20 | 65.16 | **45.12** | 54.44 | 54.00 | **45.00** | 52.44 | 62.24 | **47.71** |
| ClinTox | LIPO | AUC | 52.40 | 52.19 | **58.60** | 48.63 | 46.04 | **53.09** | 62.17 | 61.99 | **65.13** |
| | | AUPR | 45.15 | 48.78 | **52.97** | 42.73 | 43.97 | **50.41** | 57.49 | 55.69 | **62.41** |
| | | FPR95 | 92.09 | 93.07 | **82.88** | 90.46 | 93.92 | **88.42** | 79.28 | 75.70 | **70.72** |
| BBBP | BACE | AUC | 76.88 | 75.35 | **78.61** | 78.18 | 78.16 | **78.51** | 79.72 | 79.35 | **81.97** |
| | | AUPR | 68.33 | 67.83 | **72.00** | **69.44** | 69.37 | 69.34 | 72.18 | 71.59 | **79.65** |
| | | FPR95 | 68.44 | 68.06 | **65.98** | 65.20 | 63.13 | **62.85** | 59.80 | 54.31 | **53.61** |
| Esol | MUV | AUC | 84.43 | 83.98 | **85.59** | 80.61 | 79.67 | **82.41** | 87.43 | 86.86 | **89.68** |
| | | AUPR | 78.80 | 78.62 | **81.68** | 72.53 | 72.36 | **75.19** | 86.81 | 86.70 | **90.74** |
| | | FPR95 | 46.90 | 50.65 | **44.63** | 51.43 | 55.03 | **47.72** | 65.19 | 58.99 | **48.09** |
| AIDS | DHFR | AUC | 95.41 | 95.04 | **96.04** | 93.51 | 93.42 | **96.28** | 96.06 | 96.02 | **98.17** |
| | | AUPR | 91.32 | 93.10 | **95.20** | 90.24 | 91.81 | **94.78** | 89.96 | 95.82 | **98.07** |
| | | FPR95 | **12.53** | 13.73 | 15.67 | 22.85 | 23.39 | **13.83** | **6.83** | 12.11 | 7.08 |
| ENZYMES | PROTEIN | AUC | 54.69 | 54.55 | **57.81** | 57.60 | 57.75 | **60.42** | 58.73 | 56.78 | **61.53** |
| | | AUPR | 60.34 | 63.44 | **67.93** | 63.40 | 63.00 | **66.57** | 61.89 | 64.15 | **68.98** |
| | | FPR95 | 97.06 | 97.22 | **93.53** | 95.00 | 94.31 | **93.02** | 92.78 | 93.82 | **92.50** |

## 5.1 Experimental Setting

*5.1.1 Datasets.* According to previous works [24], we use six pairs of graph datasets as ID and OOD data respectively, i.e., **IMDB-M & IMDB-B**, **ClinTox & LIPO**, **BBBP & BACE**, **Esol & MUV**, **AIDS & DHFR**, and **ENZYMES & PROTEIN**. The descriptions of these dataset pairs are provided in Appendix A.4. Each dataset pair comes from the same domain and has a moderate domain shift between them. For each dataset pair, we treat the first dataset as the ID dataset and the second dataset as the OOD dataset. 60% of and 10% of graphs from the ID dataset are used for training and calibration respectively, i.e., $\mathcal{D}_{ptr}$ and $\mathcal{D}_{cal}$. The remaining 30% of ID graphs and the same number of OOD graphs from the second dataset are used for constructing $\mathcal{D}_{te}$. $\mathcal{D}_{cal}$ is further evenly divided into $\mathcal{D}_{cal-tr}$ and $\mathcal{D}'_{cal}$. $\mathcal{D}_{ptr}$ is used to derive a pre-trained OOD detection model.

*5.1.2 Baselines.* To fully evaluate our method, we compare CGOD with the following three compared categories:

- **Supervised method with detector**. This branch owns two stages. At the first stage, it would train some powerful graph encoders, such as graph kernels and GNNs in a supervised manner. Based on the learned graph representations, these methods adopt an OOD or anomaly detector to identify OOD graphs at the second stage.
- **Unsupervised method with detector**. This branch also owns two stages. The only difference is that this branch adopts the unsupervised or self-supervised manner to train graph encoders. Hence, graph contrastive learning models, such as InfoGraph [34] and GraphCL [46] can be well used at the first stage.
- **End-to-end method**. Current state-of-the-art (SOTA) models belong to this branch, e.g., GOOD-D [24]. In addition,

some promising graph anomaly detection methods, such as GLocalKD [25] are also end-to-end methods.

The description of these used baselines is provided in Appendix A.5.

*5.1.3 Implementation.* There are four main hyper-parameters in our model: the learning rate for training the proposed adaptive data augmentation, i.e., lr, the number of data augmentations, i.e., k, the discretization ratio used in Eq.(8), i.e., $\xi$, and the harmonic factor used in Eq.(9) i.e., $\lambda$. We adopt the grid search to find their optimal values. The search intervals of these hyper-parameters are reported in Appendix A.6. For a fair comparison, all baselines and CGOD share the same hidden dimensional size. In addition, since $\mathcal{D}_{cal}$ is not used in all baselines, we integrate $\mathcal{D}_{cal}$ into the training dataset for their model training.

*5.1.4 Evaluation Metrics.* Following previous works [10, 43], we adopt four OOD detection metrics: **AUC**, **AUPR**, **FPR95** and **FPR**. The first three metrics are used for evaluating detection results. A higher AUC, AUPR and a lower FPR95 indicate better model performance. FPR is used for testing whether our method can effectively control FPR within a user-specified threshold $\epsilon$. The detailed descriptions of these metrics are provided in Appendix A.7.

## 5.2 Model Adaptation (A1)

In this section, we test whether CGOD can be armed with different detection models to achieve prediction. Three representative methods from the above categories in Section 5.1.2 are selected: GIN, InfoGraph and GOOD-D. GIN and InfoGraph are classic GNNs in supervised and unsupervised settings. We arm them with SSD to achieve detection which are denoted as $GIN_D$ and $IG_D$. GOOD-D is a classic end-to-end detection model. We denote these methods armed with CGOD as $CGOD_S$, $CGOD_U$ and $CGOD_E$, respectively. We also combine them with the traditional CP framework, and

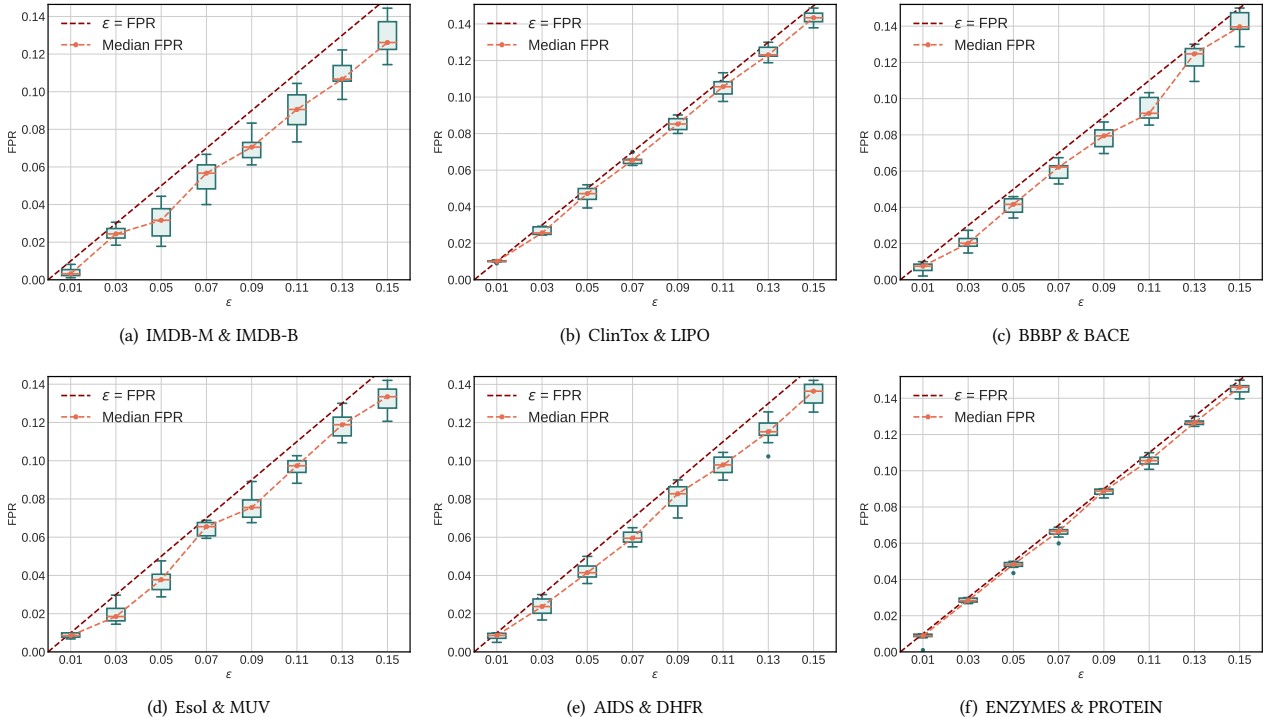

Figure 2: In our method, the FPR is upper bounded by $\epsilon$ on six ID & OOD datasets.

"+CP" is added after the model name. Table 2 shows their empirical results. The best performance is in boldface. From it, we have the following conclusions:

- CGOD can further improve the performance of different detection methods. Compared with three detection methods, the most obvious improvement is that $CGOD_U$ reduces FPR95 by 39.47% on AIDS & DHFR.
- The comparison between CGOD and those armed with the traditional CP framework demonstrates the effectiveness of our proposed aggregated non-conformity score function. The most obvious improvement is that $CGOD_U$ reduces FPR95 by 40.87% on AIDS & DHFR.
- Through optimizing the metrics of score consistency and representation diversity, our proposed adaptive data augmentation can better capture the intrinsic features among ID graphs. Therefore, CGOD can not only control the type I error rate, but also reduce the type II error rate and improve the detection performance.

## 5.3 FPR Controlling (A2)

In this section, we investigate whether the FPR can be controlled by CGOD. The user-specified threshold $\epsilon$ is selected from 0.01 to 0.15 with the step size 0.02. For each $\epsilon$, we perform 10 random splits of $\mathcal{D}_{cal-tr}$ and $\mathcal{D}'_{cal}$. We select $CGOD_E$ as the test model. The empirical results of all datasets are shown in Figure 2. Besides the theoretical result of Theorem 4.2, Figure 2 shows that the FPR can be empirically upper bounded by $\epsilon$ for all datasets in our method.

Table 3: Ablation study on six ID & OOD datasets w.r.t. AUC.

| Linear | $\mathcal{L}_s$ | $\mathcal{L}_r$ | IMDB-M IMDB-B | ClinTox LIPO | BBBP BACE | Esol MUV | AIDS DHFR | ENZYMES PROTEIN |
|---|---|---|---|---|---|---|---|---|
| ✓ | ✓ | - | 77.39 | 62.85 | 80.07 | 87.49 | 96.21 | 58.20 |
| ✓ | — | ✓ | 76.08 | 60.62 | 78.43 | 85.18 | 95.51 | 56.33 |
| ✓ | ✓ | ✓ | 78.53 | **67.08** | 81.26 | 88.12 | 96.44 | 59.88 |
| - | ✓ | ✓ | **79.02** | 65.13 | **81.97** | **89.68** | **98.17** | **61.53** |

Moreover, for different datasets, the bounded difficulty is different. The upper bounds in the sub-figures (a), (c), (d) and (e) are more relaxed than those in the sub-figures (b) and (f).

## 5.4 Performance Comparison (A3)

To further validate the effectiveness of CGOD, we also compare $CGOD_E$ with eight SOTA graph-level OOD detection models which are described in Appendix A.5. We conduct experiments on all datasets and select AUC as the evaluation metric. The empirical results are provided in Figure 3. From it, we can conclude that $CGOD_E$ consistently achieves the best detection performance on all datasets. Particularly, the most significant improvement is 4.76% increase in AUC of $CGOD_E$ compared to the SOTA baseline on ClinTox & LIPO. It is a non-trivial improvement because $CGOD_E$ uses 10% less training data. It also demonstrates the effectiveness of our framework with the proposed aggregated non-conformity score function.

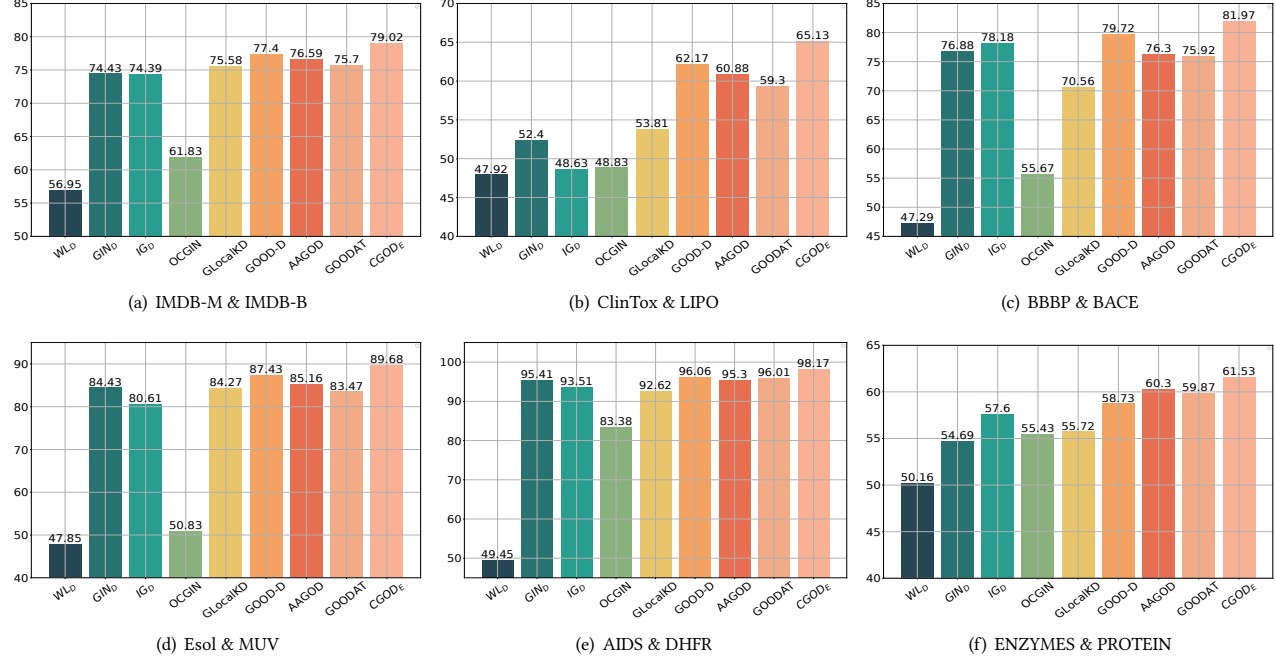

Figure 3: Performance (%) comparison on six ID & OOD datasets w.r.t. AUC.

## 5.5 Ablation Study (A4)

In this section, we perform the ablation study to valid the effectiveness of different components in CGOD. Specifically, we consider three model variants: 1) the one uses the linear operation to generate the weighted vector $a_j^i$ instead of the attention mechanism in Eq.(7); 2) the one only uses the loss function of score consistency, i.e., $\mathcal{L}_s$; 3) the one only uses the loss function of representation diversity, i.e., $\mathcal{L}_r$. We select $\text{CGOD}_E$ as the backbone, and we denote three model variants as $\text{CGOD}_E$-ls, $\text{CGOD}_E$-lr and $\text{CGOD}_E$-lsr sequentially.

The empirical results are shown in Table 3. From it, we can see that among three model variants $\text{CGOD}_E$-lsr achieves the best performance. But $\text{CGOD}_E$-lsr is still inferior to $\text{CGOD}_E$, which illustrates the effectiveness of the attention mechanism in our data augmentation. In addition, compared with $\text{CGOD}_E$ and $\text{CGOD}_E$-lsr, the performance of $\text{CGOD}_E$-ls and $\text{CGOD}_E$-lr has significantly declined. For $\text{CGOD}_E$-ls, there may be redundant features among the augmented graph representations without $\mathcal{L}_r$. For $\text{CGOD}_E$-lr, the augmented graph representations may deviate too far from the original graph representation, resulting in the introduction of noisy information in model prediction.

## 5.6 Hyper-parameter Sensibility (A5)

Figure 4 shows the hyper-parameter sensitivity of CGOD in terms of four hyper-parameters as mentioned in Section 5.1.3. We select $\text{CGOD}_E$ as the test case. From it, we have the following observations: A reasonable hyper-parameter setup is to set the learning rate and $\lambda$ to the smaller values for better designing and optimizing the loss function; $k$ and $\xi$ are suitable to assign the larger values for

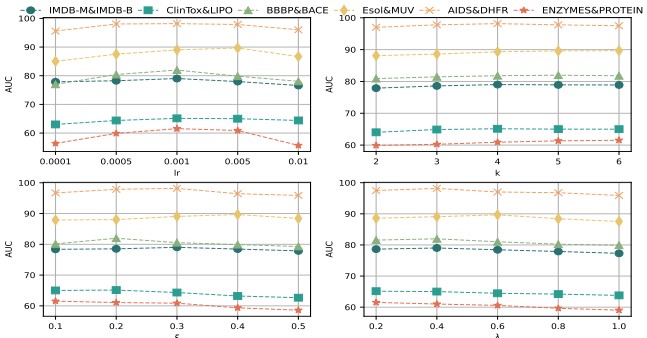

Figure 4: Hyper-parameter sensitivity of CGOD.

improving model expressiveness, but setting them to an excessively large value can also increase the risk of overfitting.

## 6 CONCLUSION

In this paper, we present the CGOD, a novel framework that extends the theory of CP to graph-level OOD detection for providing a rigorous control over detection results. CGOD differentiates the process of conformal inference and proposes a trainable augmentation strategy, i.e., adaptive data augmentation to generate multiple non-conformity scores. Building on this, CGOD further introduces a new aggregated non-conformity score function that can aggregate these generated scores to improve the robustness and accuracy of the non-conformity score estimation. Extensive experiments show that CGOD outperforms many strong baselines and guarantees a bounded FPR effectively.

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

## A APPENDIX

### A.1 Proof of Theorem 4.1

Given a probability space $(\Omega, \mathcal{F}, \mathbb{P})$, the set of graph data augmentations $\mathcal{S}$ is a subset of all augmentations on the feature space $\mathcal{X}$. Both $\mathcal{S}$ and $\mathcal{X}$ are equipped with respective $\sigma$-algebras $\mathcal{F}_{\mathcal{S}}$ and $\mathcal{F}_{\mathcal{X}}$. Sampling a random data augmentation $\Psi \in \mathcal{S}$ corresponds to a measurable function $\widetilde{\Psi} : \mathcal{X} \times \Omega \to \mathcal{S}$, which allocates an element $\omega \in \Omega$ to a specific graph data augmentation $\Psi = \widetilde{\Psi}(\cdot; \omega)$. The distribution of $\widetilde{\Psi}(\cdot; \omega)$ denoted as $\mathbb{P}^{\Psi}$ is a probability measure on $(\mathcal{S}, \mathcal{F}_{\mathcal{S}})$. Based on this, we can construct the product measure for sampling i.i.d data augmentations for graphs.

First, we have that the proper training set $\mathcal{D}_{ptr}$ and the calibration set $\mathcal{D}'_{cal}$ are i.i.d:

$$\mathcal{D}_{ptr} \overset{\text{i.i.d}}{\sim} \mathcal{D}'_{cal} \overset{\text{i.i.d}}{\sim} \mathbb{P}^{in}, \tag{15}$$

which follow the same distribution $\mathbb{P}^{in}$. When the test graph $G_{te}$ is from $\mathbb{P}^{in}$, we then have that

$$(G_{te}, \Psi^{te,1:k}) \overset{\text{i.i.d}}{\sim} \{(G_j^{in}, \Psi^{j,1:k}) | G_j^{in} \in \mathcal{D}'_{cal}\}_{j=n'+1}^n \overset{\text{i.i.d}}{\sim} \mathbb{P}^{in} \times \mathbb{P}^{\Psi}. \tag{16}$$

Meanwhile, the aggregated non-conformity score function $\widehat{V}(\cdot, \cdot; \cdot)$ relies on $\mathcal{D}_{ptr}$, the transformation function $F(\cdot, \cdot)$ in Eq.(2), and the sampled data augmentations, i.e., $\Psi^{te,1:k} \cup \{\Psi^{j,1:k}\}_{j=n'+1}^n$. Hence, the generated $k$ non-conformity scores $s_{te}^{1:k}$ of $G_{te}$ and those for the graphs in $\mathcal{D}'_{cal}$, i.e., $\{s_j^{1:k}\}_{j=n'+1}^n$ are also i.i.d conditioned on $\widehat{V}(\cdot, \cdot; \cdot)$, $\mathcal{D}_{ptr}$ and $\Psi^{te,1:k} \cup \{\Psi^{j,1:k}\}_{j=n'+1}^n$.

Due to the fact that $\widehat{V}(\cdot, \cdot; \cdot)$ is an aggregation function of the generated non-conformity scores for guaranteeing that it is an element-wise strictly increasing function, so these $n - n' + 1$ aggregated non-conformity scores $(\hat{s}_{te}, \hat{s}_{n'+1}, \ldots, \hat{s}_n)$ are also i.i.d:

$$\hat{s}_{te} \overset{\text{i.i.d}}{\sim} \hat{s}_{n'+1} \overset{\text{i.i.d}}{\sim} \ldots \overset{\text{i.i.d}}{\sim} \hat{s}_n. \tag{17}$$

Following the above formulation, we can regard these aggregated non-conformity scores $(\hat{s}_{te}, \hat{s}_{n'+1}, \ldots, \hat{s}_n)$ as i.i.d $n - n' + 1$ random variables with their corresponding continuous densities. Summing up the above, we can derive that

$$|\{\hat{s}_{te} \leq \hat{s}_j \mid j = n' + 1, \ldots, n\}| \sim \text{Uniform}(\{1, \ldots, n - n' + 1\}). \tag{18}$$

Therefore, the aggregated $p$-value in Eq.(12), i.e., $\hat{p}_{te}$ is uniformly distributed over $\{\frac{1}{n-n'+1}, \frac{2}{n-n'+1}, \ldots, 1\}$, and this proof is completed. In addition, besides the strictly increasing constraint, we can also add a small amount of random noise to these aggregated non-conformity scores.

### A.2 Proof of Theorem 4.2

The independence between $\mathcal{D}'_{cal}$, $\mathcal{D}_{te}$ and $\widehat{V}(\cdot, \cdot; \cdot)$ is easily to be satisfied. If the test graph $G_{te}$ is sampled from $\mathbb{P}^{in}$, then its corresponding aggregated $p$-value $\hat{p}_{te}$ follows a discrete uniform distribution based on Theorem 4.1, i.e., $\hat{p}_{te}$ is uniformly distributed over $\{\frac{1}{n-n'+1}, \frac{2}{n-n'+1}, \ldots, 1\}$. The probability of the event that $G_{te} \sim \mathbb{P}^{in}$

---

**Algorithm 1:** Training and inference procedures of CGOD.

1: **#Training Procedure:**
2: **Input:** training set $\mathcal{D}_{tr}$.
3: **Output:** learned adaptive data augmentations.
4: Split $\mathcal{D}_{tr}$ into $\mathcal{D}_{ptr}$ and $\mathcal{D}_{cal}$, and further split $\mathcal{D}_{cal}$ into $\mathcal{D}_{cal-tr}$ and $\mathcal{D}'_{cal}$.
5: Pre-train an OOD detection model on $\mathcal{D}_{ptr}$.
6: **while** *not done* **do**
7:   **for** randomly sampling a graph $G_j^{in} \in \mathcal{D}_{cal-tr}$ **do**
8:     **for** each data augmentation $\Psi^{j,i} \in \Psi^{j,1:k}$ **do**
9:       Generate the augmented representation $\hat{g}_j^i$ by Eq.(7-8).
10:       Calculate its non-conformity score $s_j^i$ by Eq.(3).
11:     **end for**
12:     Calculate the loss function $\mathcal{L}$ in Eq.(9), and update model parameters by Adam optimizer.
13:   **end for**
14: **end while**
15: **#Inference Procedure:**
16: **Input:** test dataset $\mathcal{D}_{te}$; learned adaptive data augmentations.
17: **Output:** detection results of $\mathcal{D}_{te}$.
18: **for** each test graph $G_{te} \in \mathcal{D}_{te}$ **do**
19:   **for** each data augmentation $\Psi^{te,i} \in \Psi^{te,1:k}$ **do**
20:     Generate the augmented representation $\hat{g}_{te}^i$ by Eq.(7-8).
21:     Calculate its non-conformity score $s_{te}^i$ by Eq.(3).
22:   **end for**
23:   Calculate its aggregated non-conformity score $\hat{s}_{te}$ by Eq.(6).
24:   Derive its $p$-value $\hat{p}_{te}$ with $\mathcal{D}'_{cal}$ by Eq.(12).
25:   If $\mathbb{I}(\hat{p}_{te} < \epsilon)$: $G_{te}$ is an OOD graph; otherwise, it is not.
26: **end for**

---

and $\hat{p}_{te} < \epsilon$ is

$$\begin{aligned}\Pr(\hat{p}_{te} < \epsilon | G_{te} \sim \mathbb{P}^{in}) &= \sum_{j=1}^{(n-n'+1)\epsilon} \frac{1}{n - n' + 1} \\ &= \frac{\lfloor (n - n' + 1)\epsilon \rfloor}{n - n' + 1} \\ &\leq \epsilon.\end{aligned} \tag{19}$$

If $\hat{p}_{te}$ is smaller than $\epsilon$, then $G_{te}$ is recognized as the OOD graph. Hence, if $G_{te} \sim \mathbb{P}^{in}$, the probability of the event that CGOD falsely recognizes $G_{te}$ as the OOD graph is upper bounded by $\epsilon$.

### A.3 Pseudo-code Description

Algorithm 1 illusrates the training and inference procedures of CGOD. From it, we can see that our model follows a simple computational process.

### A.4 Dataset Description

We adopt six pairs of real-world datasets from three domains:

- Social networks datasets: **IMDB-M** & **IMDB-B**. They are movie collaboration datasets, where nodes denote actors and an edge is drawn between two actors if they appear in the same movie.

- Molecule datasets: **ClinTox** & **LIPO**, **BBBP** & **BACE**, **Esol** & **MUV**, and **AIDS** & **DHFR**. For each dataset, nodes denote atoms and an edge connecting two nodes is a chemical bond of molecule graphs.
- Bioinformatics datasets: **ENZYMES** & **PROTEIN**. They are macromolecule datasets, where nodes denote secondary structure elements and an edge represents that two nodes are neighbors along the amino acid sequence or one of three nearest neighbors in space.

## A.5 Baseline Description

We compare our method with the following strong baselines:

- **WL** [32] represents the Weisfeiler-lehman graph kernel, which processes a graph by re-labeling each node with a new label compressed from a multiset label consisting of its original label and the sorted labels of its neighbors.
- **GIN** [45] is a representative work of supervised GNNs, which designs the new neighbor aggregation and readout functions to improve GNN expressiveness.
- **InfoGraph** [34] is a representative work of unsupervised GNNs. It designs a self-supervised objective to maximize the mutual information between the graph-level representation and the sub-structure representations.
- **OCGIN** [49] is a one-class anomaly detection model for recognizing abnormal graphs, which adopts GIN and SVDD (support vector data description) as the encoder and the objective at the output layer, respectively.
- **GLocalKD** [25] is a popular method of graph-level anomaly detection. It performs the joint random distillation of graph and node representations to capture local- and global-patterns to detect anomalous graphs.
- **GOOD-D** [24] is the first work to solve graph-level OOD detection, which proposes a hierarchical graph contrastive learning framework to identify OOD graphs via different granularity scores.
- **AAGOD** [10] is a post-hoc method which designs an amplifier as prompts for recognizing the key information from the graph structure, enabling a well-trained GNN to achieve OOD detection.
- **GOODAT** [38] considers how to achieve graph-level OOD detection at the test time. It leverages the information bottleneck to learn informative subgraphs for enlarging the gap between ID and OOD graphs.

We use the implementations of GIN and InfoGraph in a well-known GNN library[3]. For WL[4], OCGIN[5], GLocalKD[6], GOOD-D[7], AAGOD[8] and GOODAT[9], we adopt their public implementations and adapt them into our training and inference pipelines. According to the relevant literature, we respectively arm WL, GIN and InfoGraph with the local outlier factor (LOF), SSD and SSD as the transformation function.

---

[3]https://www.pyg.org/
[4]https://ysig.github.io/GraKeL/0.1a8/
[5]https://github.com/LingxiaoShawn/GLOD-Issues
[6]https://github.com/RongrongMa/GLocalKD
[7]https://github.com/yixinliu233/G-OOD-D
[8]https://github.com/BUPT-GAMMA/AAGOD
[9]https://github.com/Ee1s/GOODAT

**Table 4: Search intervals of hyper-parameters.**

| Hyper-parameter | Search Interval |
|---|---|
| learning rate lr | {0.0001, 0.0005, 0.001, 0.005, 0.01} |
| augmentation number $k$ | {2, 3, 4, 5, 6} |
| discretization ratio $\xi$ | {0.1, 0.2, 0.3, 0.4, 0.5} |
| harmonic factor $\lambda$ | {0.2, 0.4, 0.6, 0.8, 1.0} |

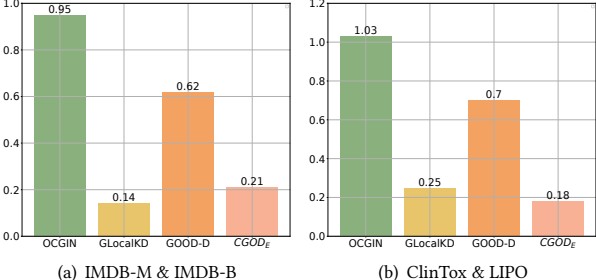

(a) IMDB-M & IMDB-B    (b) ClinTox & LIPO

**Figure 5: Training time comparison (in seconds).**

## A.6 Hyper-parameter Setting

As described in Section 5.1.3, there are four hyper-parameters in our method: lr, $k$, $\xi$ and $\lambda$. Their corresponding search intervals are given in Table 4.

## A.7 Metric Description

- **AUC** refers to area under the receiver operating characteristic curve, which summarizes the ROC curve into a single value to demonstrate the model performance with different thresholds.
- **AUPR** refers to area under the precision-recall curve. Compared with AUC, AUPR can adjust different positive and negative class rates to alleviate the problem of imbalanced class.
- **FPR95** is the false positive rate at 95% true positive rate. It is a commonly used metric to evaluate the performance of classification models, especially in the tasks of binary classification and anomaly or OOD detection.
- **FPR** is a statistical measure that represents the proportion of actual negative samples that are incorrectly classified as positive by a detection model.

## A.8 Runtime Comparison

We compare the training runtime between $CGOD_E$ and three end-to-end detection methods including OCGIN, GLocalKD and GOOD-D in an epoch. The experiments are conducted on a Linux server with Intel(R) Xeon(R) CPU E5-2620 v4 @ 2.10GHz, 128G RAM and NVIDIA Tesla V100. IMDB-M & IMDB-B and ClinTox & LIPO are selected datasets. Figure 5 shows the concrete results. From it, we conclude that CGOD keeps a good training efficiency.

