# OpenReview forum: "Conformal Graph-level Out-of-distribution Detection with Adaptive Data Augmentation"
_ACM.org/TheWebConf/2025/Conference — WWW 2025 Poster_

### Official Review · Reviewer_gmvS · 2024-11-05

**Novelty:** 5
**Technical Quality:** 4

**Review:**

This paper introduces a graph-level out-of-distribution (OOD) detection framework that extends conformal prediction (CP) to the context of graph-level analysis, ensuring rigorous control over the false positive rate. The framework employs adaptive data augmentation using an attention-based mechanism to perturb graph representations within the embedding space. It also integrates two key metrics—score consistency and representation diversity—to enhance the robustness of the OOD detection process.

Pros:

1. The proposed framework effectively addresses the limitations of prior approaches that lack statistical guarantees for detection results in graph-level OOD detection.

2. An innovative aggregated non-conformity score function is introduced to improve the robustness of the detection process.

3. Both the theoretical and experimental analyses are comprehensive and thorough.

Cons:

1. The paper would benefit from a dedicated section discussing its limitations and outlining potential directions for future work.

2. The linear operation discussed in Section 5.5 lacks a clear explanation.

**Questions:**

Please kindly refer to the weaknesses.

**Reviewer Confidence:**

2: The reviewer is willing to defend the evaluation, but it is likely that the reviewer did not understand parts of the paper

**Scope:**

4: The work is relevant to the Web and to the track, and is of broad interest to the community

---

### Official Review · Reviewer_bjCb · 2024-11-29

**Novelty:** 6
**Technical Quality:** 4

**Review:**

**Summary:**
The paper introduces an aggregated non-conformity score for detecting out-of-distribution graphs and validates it by comparing against eight baseline methods on six dataset pairs (each pair comprising in-distribution and out-of-distribution datasets). The topic is relevant and aligns with the scope of the conference. However, there are areas where clarity, additional experiments, and corrections are needed to fully showcase the research's value.

**Suggestions for Improvement:**
**Paper Length and Depth:**
Consider submitting to a venue that allows longer papers to include additional clarifications, experiments, and annexes, enabling a more comprehensive presentation of the work.
**Dataset Selection and Results:**
Reference \[24\] includes 10 dataset pairs, but the paper only uses 6 pairs. Provide reasoning for this selection and discuss whether results are consistent across all 10 pairs.
Including experiments on the remaining dataset pairs would strengthen the validation of the approach.
**Proposed Metrics Validation:**
Evaluate whether the proposed metrics are inherently biased toward the proposed model. Discuss their generalizability and potential biases.
**Ablation Study (Section 5.5):**
The ablation study is difficult to interpret, probably due to space restrictions. Clarify which models correspond to the rows in Table 3\.
Specify whether a tick in Table 3 indicates the inclusion or exclusion of a component.
Clearly state in the text whether CGODE is the full model and how it maps to Table 3 (row 3 or 4).
**Terminology and Model Notation:**
Clearly explain CGOD with sub-indexes 𝑆, 𝐷 and 𝐸 in the text to avoid confusion.
**Baselines and Datasets:**
Describe all baseline algorithms and datasets used in the study within the main text, not just in the annex. This ensures the paper is self-contained.
**Formatting Issues:**
Correct the misassigned bold values in Table 2\.

**Strengths:**
The paper addresses a relevant problem aligned with the conference scope.
The proposed method shows promise in detecting out-of-distribution graphs.
**Weaknesses:**
Lack of clarity in key sections (e.g., ablation study, model notation).
Limited dataset coverage and absence of discussion on excluded datasets.
Insufficient validation of proposed metrics' generalizability.

**Questions:**

Have you conducted experiments on all 10 dataset pairs mentioned in reference [24]? If yes, please summarize the results and explain why only 6 pairs are included in the paper.

Could your proposed metrics be biased toward your method? If not, please provide evidence or reasoning to address this concern.

Can you clearly explain the ablation study results in plain text, including the role of each component?

**Reviewer Confidence:**

3: The reviewer is confident but not certain that the evaluation is correct

**Scope:**

4: The work is relevant to the Web and to the track, and is of broad interest to the community

---

### Official Review · Reviewer_WoiR · 2024-11-30

**Novelty:** 4
**Technical Quality:** 4

**Review:**

This paper presents CGOD, a novel framework that extends conformal prediction to graph-level out-of-distribution (OOD) detection, providing statistically guaranteed results. CGOD introduces adaptive data augmentation to generate multiple non-conformity scores, which are then aggregated to enhance robustness. Experiments on multiple real-world datasets demonstrate its superior performance over competitive baselines, particularly in controlling the false positive rate.

Strength:
- Statistical Guarantees: CGOD provides statistical guarantees on the false positive rate (FPR) through the extension of conformal prediction to graph-level OOD detection.
- Adaptive Data Augmentation: The introduction of adaptive data augmentation generates multiple non-conformity scores, enhancing the detection's robustness and accuracy.
- Comprehensive Experimental Evaluation: The paper evaluates CGOD on multiple real-world datasets, demonstrating its superior performance and robustness in various scenarios.

Weaknesses:
- Assumptions and Exchangeability Properties: The paper assumes the exchangeability of graph data but does not fully verify whether this property is maintained through data augmentation. The i.i.d. assumption before and after augmentation is not explicitly checked. This may affect the i.i.d. property and statistical guarantees.
- Experimental Validation and Generalizability: The current experiments focus on a narrow range of OOD data. To test generalizability, the paper should use multiple combinations of OOD data. However, OOD data may not have the exchangeability or i.i.d. property, which could affect the model's performance and theoretical rigor. This is a significant limitation and is important to consider for both the theoretical and practical boundaries of the model.
- Lack of Some Related Works: The authors mainly focus on using multiple data augmentation to construct a new nonconformity measurement. This ensemble strategy may be related to the uncertainty awareness strategy (Uncertainty-Aware Conformalized Quantile Regression). The paper should give a more comprehensive review of this sort of research and distinguish the differences in nonconformity measurements to provide a clearer understanding of the unique contributions of the proposed method.

**Questions:**

Please reply to the weaknesses if I have some misunderstanding. Thanks.

**Reviewer Confidence:**

3: The reviewer is confident but not certain that the evaluation is correct

**Scope:**

3: The work is somewhat relevant to the Web and to the track, and is of narrow interest to a sub-community

---

### Official Review · Reviewer_GMrr · 2024-12-04

**Novelty:** 4
**Technical Quality:** 6

**Review:**

**1. Summary**

This paper introduces CGOD, a framework for graph-level out-of-distribution (OOD) detection that ensures statistical guarantees for detection results. To achieve this, the authors extend conformal prediction theory to the graph domain and propose an adaptive non-conformity scoring mechanism powered by data augmentation. Experiments across six dataset pairs demonstrate that CGOD outperforms existing methods in both accuracy and reliability, showcasing its practical utility in safety-critical applications.

**2. Pros**

(P1) Important problem: The paper addresses a critical and relevant problem in graph-level OOD detection

(P2) Versatile methodology: The proposed methodology is versatile and can integrate with various detection models.

(P3) Theoretical analysis on detection guarantee: Theoretical analysis provides strong support for the statistical guarantees of the proposed method.


**3. Cons**

(C1) Better justification of practical significance for statistical guarantees

While the paper provides theoretical and empirical evidence of FPR control, the practical significance of this guarantee in real-world applications remains underexplored. Additionally, the observed FPR improvements over state-of-the-art methods (e.g., GOOD-D) are not consistently significant across datasets, requiring a deeper discussion of the specific contexts where CGOD offers unique advantages.

(C2) Lack of dataset statistics

The authors do not provide detailed statistics (e.g., number of nodes, graphs, and edges) for the datasets used, making it difficult to evaluate the scalability and generalizability of the proposed method in diverse real-world scenarios. Including these details would strengthen the experimental evaluation.

**Questions:**

Please refer to the Review.

**Reviewer Confidence:**

2: The reviewer is willing to defend the evaluation, but it is likely that the reviewer did not understand parts of the paper

**Scope:**

3: The work is somewhat relevant to the Web and to the track, and is of narrow interest to a sub-community